# FluB-RAM and FluB-RANS: Genome Rearrangement as Safe and Efficacious Live Attenuated Influenza B Virus Vaccines

**DOI:** 10.3390/vaccines9080897

**Published:** 2021-08-12

**Authors:** Stivalis Cardenas-Garcia, C. Joaquín Cáceres, Aarti Jain, Ginger Geiger, Jong-Suk Mo, Algimantas Jasinskas, Rie Nakajima, Daniela S. Rajao, D. Huw Davies, Daniel R. Perez

**Affiliations:** 1Department of Population Health, College of Veterinary Medicine, University of Georgia, Athens, GA 30602, USA; stivalis@uga.edu (S.C.-G.); cjoaquincaceres@uga.edu (C.J.C.); imginger@uga.edu (G.G.); jm45001@uga.edu (J.-S.M.); daniela.rajao@uga.edu (D.S.R.); 2Department of Physiology and Biophysics, School of Medicine, University of California Irvine, Irvine, CA 92697, USA; aartij@hs.uci.edu (A.J.); ajasinsk@hs.uci.edu (A.J.); rie3@hs.uci.edu (R.N.); ddavies@uci.edu (D.H.D.)

**Keywords:** LAIV, influenza, HA, IgA, IgG, vaccine, genome rearrangement

## Abstract

Influenza B virus (IBV) is considered a major respiratory pathogen responsible for seasonal respiratory disease in humans, particularly severe in children and the elderly. Seasonal influenza vaccination is considered the most efficient strategy to prevent and control IBV infections. Live attenuated influenza virus vaccines (LAIVs) are thought to induce both humoral and cellular immune responses by mimicking a natural infection, but their effectiveness has recently come into question. Thus, the opportunity exists to find alternative approaches to improve overall influenza vaccine effectiveness. Two alternative IBV backbones were developed with rearranged genomes, rearranged M (FluB-RAM) and a rearranged NS (FluB-RANS). Both rearranged viruses showed temperature sensitivity in vitro compared with the WT type B/Bris strain, were genetically stable over multiple passages in embryonated chicken eggs and were attenuated in vivo in mice. In a prime-boost regime in naïve mice, both rearranged viruses induced antibodies against HA with hemagglutination inhibition titers considered of protective value. In addition, antibodies against NA and NP were readily detected with potential protective value. Upon lethal IBV challenge, mice previously vaccinated with either FluB-RAM or FluB-RANS were completely protected against clinical disease and mortality. In conclusion, genome re-arrangement renders efficacious LAIV candidates to protect mice against IBV.

## 1. Introduction

Influenza B viruses (IBVs) in the *Orthomyxoviridae* family were first isolated in 1940 in Irvington, NY [1]. IBVs are enveloped by a host-derived lipid bilayer and contain eight segments of single-stranded, negative-sense RNA [2] that encode for at least 11 proteins: polymerase basic 1 (PB1), polymerase basic 2 (PB2), polymerase acidic (PA), hemagglutinin (HA, surface glycoprotein), nucleoprotein (NP), neuraminidase (NA, surface glycoprotein), NB (surface glycoprotein), matrix protein 1 (M1), matrix protein 2 (BM2), non-structural protein 1 (NS1), and non-structural protein 2 (NS2) [3,4,5,6,7]. IBVs are of public health relevance thanks to their association with severe respiratory disease in humans, particularly in pediatric and elderly populations. Two antigenically distinct lineages co-circulate worldwide identified as Victoria and Yamagata lineages that show no serological cross reactivity, providing limited cross protection against each other [8,9,10]. The incidence of IBV infections varies from season to season, linked to 0.69–61% of the influenza-induced pediatric mortalities registered in the United States from 2004 to 2020 [11]. During the 2019–2020 influenza season, IBV showed an early onset and the incidence of IBV infections in the United States increased compared with previous seasons. Compared with the 2018–2019 influenza season in which about 7% of influenza-positive samples corresponded to IBV [12], >45% of the influenza-positive samples were positive for IBV during the 2019–2020 season. The 2019–2020 IBV season was associated with 122 (61.3%) of pediatric deaths [11] and 5174 (26.8%) hospitalizations, with the highest rate among adults ≥65 years old [13].

Although vaccination is the most effective strategy to ameliorate the impact of influenza infections, the incidence of IBV shows an increasing trend. This is in part due to vaccine mismatch in trivalent vaccine formulations that contain only one IBV strain from one of the lineages [14,15,16,17,18,19,20,21]. These observations underscore the importance of including both IBV lineages in seasonal vaccine formulations, as is the case in several of the most recently FDA-approved quadrivalent vaccines [22]. However, additional efforts are warranted in order to improve vaccine protection against IBV. Live attenuated vaccine platforms have been among the most explored over the years (reviewed in [23]). In addition to the cold-adapted LAIVs developed in the 1960s that form the basis of the current LAIVs approved for human use, alternative LAIV approaches have been developed that include modifications and deletions to the NS1 gene segment, generation of M2 deficient viruses, and alternative virus backbones with temperature sensitive phenotypes, among others [24,25,26,27,28,29,30]. We have previously shown that genome re-arrangement is a suitable strategy for the development of influenza A virus LAIVs [31]. In the present study, we expanded these studies into IBV and produced two distinct genome re-arrangements in the backbone of the B/Brisbane/60/2008 strain (Victoria lineage). The FluB-RAM re-arrangement involved producing a chimeric segment 1 that encodes PB1 and BM2, and a series of mutations in segment 7 to completely abrogate expression of BM2 from the latter. The FluB-RANS re-arrangement used a similar strategy, whereby NS2 was cloned downstream of PB1 and segment 8 contains multiple mutations that precludes NS2 expression. The safety and efficacy of the FluB-RAM and FluB-RANS viruses were evaluated in DBA/2J mice [26,30,32,33]. Both vaccine candidates were immunogenic and effectively protected mice against homologous lethal IBV challenge.

## 2. Materials and Methods

### 2.1. Cells and Eggs

Madin Darby canine kidney (MDCK) cells and 293T cells (ATCC CRL-3216) were used for reverse genetics of virus strains. Specific pathogen-free embryonated chicken eggs (ECEs) used for virus propagation and stock titration were obtained from Charles Rivers (Wilmington, MA, USA).

### 2.2. Recombinant Plasmids

DNA fragments flanked by AarI sites and encoding, in the 5′-3′ direction, the 82 codons of the C-terminus of B/Bris PB1, followed by codons encoding the sequence Gly-Gly-Gly-Gly-Ser (G4S), the 2A protease from Thosea asigna virus ORF (Tav 2A), either the BM2 ORF or BNS2 ORF of B/Bris, followed by the untranslated region of B/Bris PB1, were synthesized and cloned into pUC57 using GenScript services (Piscataway, NJ, USA). The synthetic fragments were subcloned using appropriate restriction sites into the reverse genetics pDP2002 vector encoding the wild type B/Bris PB1 gene segment [26] to generate the plasmids pSCG_PB1G4S2ATavBM2_FluB (pSCG-PB1BM2) and pSCG_PB1G4S2ATavBNS2_FluB (pSCG-PB1BNS2), respectively. The reverse genetics plasmids that encode the B/Bris M segment and B/Bris NS segment were mutagenized to obliterate expression of BM2 and BNS2, respectively. In plasmid pSCG_M_FLuB_stops_at_BM2 (pSCG-BM1-∆M2), the nucleotide sequence 771-AGTGATCTAATGATTTCAGATTCTTACAATTTGTTCTTTTATCTTATCAGCTCTCCATTTCTAGGCTTGGACAATAGGGCATTTGAATCAAATAAAAAGAGGAATAAACTAG-881 leads to the following aa mutations: an extra stop codon at the end of the M1 ORF; and M1V, L2I, E3Stop, P4Stop, M21Stop, and M37Stop for BM2 ORF (Figure 1A). In pSCG_NS_FLuB_stops_at_NS2 (pSCG-BNS1-∆NEP), the sequence 733-CTGTAGAGGACGAAGAAGACGGCCATCGGATCCTCAACTCACTCTTCGAGCGTCTTAACGAAGGACATTCAAAGCCAATAA-813 leads to the following aa mutations: Q to L at the acceptor splicing boundary, W13Stop, M15T, M18T, M31T, and F38Stop (Figure 1A). Plasmids were propagated in the top 10 chemically competent E. coli cells (ThermoFisher Scientific, Waltham, MA, USA). Plasmid purifications were carried out using QIAGEN Plasmid Maxi Kit (Qiagen, Gaithersburg, MD, USA). The modifications on the plasmids were confirmed by Sanger sequencing using Psomagen services (Rockville, MD, USA).

### 2.3. Rescue of FluB-RAM and FluB-RANS Viruses with Rearranged Genomes

Recombinant viruses were rescued by reverse genetics as previously described [34]. We employed a 6 + 2 method whereby six plasmids, each containing a single cDNA copy of the wild type gene segments from the B/Brisbane/60/2008, were mixed with the corresponding pair of plasmids (either pSCG-PB1BM2 and pSCG-BM1-∆M2, or pSCG-PB1BNS2 and pSCG-BNS1-∆NEP) to produce the B/Bris rearranged M (FluB-RAM) and B/Bris rearranged NS (FluB-RANS) viruses, respectively. The identity of the rescued viruses was confirmed by Sanger sequencing (Psomagen). The recombinant viruses were propagated and titrated in 11-day-old SPF ECEs incubated at 33 °C for 48 h. Virus stocks were stored at −80°C until further use. These stocks constitute the first passage in ECEs (E1).

### 2.4. Stability of FluB-RAM and FluB-RANS Viruses through Serial Passages in ECEs

Serial passages were performed in 11-day-old SPF embryonated chicken eggs as follows: serial 10-fold dilutions from FluB-RAM and FluB-RANS E1 viruses were prepared in 1X phosphate buffered saline (PBS) and 100 µL from each dilution was inoculated into each of five ECEs through the allantoic cavity to generate E2. The inoculated ECEs were incubated at 33 °C for 48 h. Allantoic fluids were then tested for hemagglutination activity by the hemagglutination assay (HA). Fluids collected from the previous to the last dilution with 5/5 embryos positive for HA activity were pooled together and used to prepare 10-fold dilutions to inoculate the next set of embryos. The same procedure was repeated until five passages had been completed, generating passage E6. Aliquots from each passage were stored at −80 °C until needed. RNA was extracted from fluids collected at each passage and from the original virus stock using the MagNA Pure LC Total Nucleic Acid Isolation Kit (Roche, San Francisco, CA, USA). The PB1, M, and/or NS gene segments were amplified by RT-PCR using SuperScript III One-Step RT-PCR System with Platinum Taq DNA Polymerase (ThermoFisher Scientific). Sanger sequencing (Psomagen) was then performed from the resulting RT-PCR products to confirm the re-arrangement at the PB1 gene segments and the presence of the introduced mutations within the M and NS gene segments, respectively. Multi-segment RT-PCR (using the same RT-PCR system) was performed as previously described [35] for full genome sequencing using next generation sequencing (NGS) as follows: amplicon libraries were prepared using the Nextera XT DNA library preparation kit (Illumina, San Diego, CA, USA) following the manufacturer’s protocol. Barcoded libraries were multiplexed and sequenced on the high-throughput Illumina MiSeq NGS platform (Illumina) in a paired-end 150-nucleotide run format. De novo genome assembly was performed as described previously [36].

### 2.5. Virus Growth Kinetics

MDCK cells were seeded in six-well plates and incubated overnight at 37 °C, under 5% CO_2_. The next day, cells were inoculated with 0.01 MOI of either the B/Bris WT, FluB-RAM, or FluB-RANS virus contained in 500 µL, each in triplicate wells. Three set of plates were prepared for each virus. Inoculated cells were incubated for 1 h at 35 °C/5% CO_2_ with gentle rocking of the plates every 15 min. Subsequently, the virus inoculum was removed, and the cells were washed twice with 1× PBS and replenished with 2 mL of fresh Opti-MEM (Gibco, ThermoFisher Scientific) supplemented with 1× antibiotic/antimycotic solution (Gibco, ThermoFisher Scientific) and 1 µg/mL of L-1-tosylamido-2-phenylethyl chloromethyl ketone (TPCK)-treated Trypsin. Plates were set to incubate at either 33, 35, or 37 °C at 5% CO_2_. Supernatants (200 µL) were collected at 0, 12, 24, 48, 72, and 96 h post-inoculation (hpi) and stored at −80 °C until processed. The amount of virus present in the collected samples was titrated by TCID_50_ in MDCK cells, determining virus presence by HA assay. Virus titers were calculated using the Reed and Muench protocol [37] and plotted as the mean TCID_50_/mL ± SD.

### 2.6. Mouse Studies

Male and female DBA/2J mice (5 weeks old) were purchased from Jackson’s Laboratories (Bar Harbor, ME) and raised until 7 weeks of age. Mice were housed in negative pressure caging in the Davison Life Sciences Complex, University of Georgia and were provided food and water *ad libitum* for the duration of the experiment.

### 2.7. Vaccine Safety

A prime-boost strategy using the same administration route and inoculum was implemented 20 days apart. Seven-week-old mice were vaccinated intranasally (i.n.) with 50 µL of inoculum distributed equally between nares. Male and female mice, housed separately, were allocated into four groups (½ females/group) as follows: G1. FluB-RAM (*n* = 12); G2. FluB-RANS (*n* = 12); G3. 1× PBS (mock, *n* = 24); and G4. B/Bris WT (*n* = 12, positive control). The FluB-RAM, FluB-RANS, and control B/Bris WT viruses were administered at a target dose of 10^6^ EID_50_/mouse. Mice were monitored daily to record clinicals signs and mortality. Body weight was recorded daily for up to 12 days following vaccination (dpv) and boost (dpb). At 19 dpb, a subset of mice from each group (*n* = 4/group, ½ females) was anesthetized with isoflurane, terminally bled to collect sera, and subsequently humanely euthanized (Figure 2A).

### 2.8. Vaccine Efficacy

Mice from the vaccine safety study (*n* = 8/group, ½ females) were challenged i.n. with a lethal dose (10^7^ EID_50_/mouse) of the B/Brisbane/60/2008 PB2-F406Y (B/Bris/ F406Y) strain [26] contained in 50 µL. A subset of mice in the mock group (*n* = 8, ½ female) remained unchallenged and served as negative controls. Mice were monitored twice daily to record clinical signs and mortality for up to 14 days post-challenge (dpc). Body weight was recorded for up to 12 dpc. At 14 dpc, survivors were anesthetized, terminally bled to collect sera, and subsequently humanely euthanized (Figure 2A).

### 2.9. Hemagglutination Inhibition (HI) Assay

Sera were prepared from whole blood collected at 19 dpb (*n* = 4/group, except for FluB-RAM) and 14 dpc (n = 8/group) by centrifugation at 1000× *g* for 15 min at room temperature. The sera were treated with receptor destroying enzyme (RDE) and the HI assay was performed in V-bottomed microtiter plates, using four hemagglutination units (HAU) of viral antigen (B/Bris WT) per 25 µL, as recommended by the OIE [1], using a suspension of turkey red blood cells (0.5%). HI titers were plotted using Prism v9 (GraphPad, San Diego, CA, USA). The limit of detection was at dilution of 1/10, and samples with undetectable titers were assigned a dilution value of 1/8 for statistical purposes.

### 2.10. Virus Neutralization (VN) Assay

Sera collected at 19 dpb were treated with RDE. In 96-well plates, twofold dilutions (50 µL) were prepared from treated sera using 1X PBS supplemented with antibiotic/antimycotic solution. Next, 100 TCID_50_ (in 50 µL) of either B/Bris (homologous) or B/Wisconsin/01/2010 (B/Wis, heterologous) were added to the corresponding wells containing serum dilutions. Serum/virus mixes were incubated at 37 °C for 1 h. Thereafter, the serum/virus mixes were added to MDCK cell monolayers and set to incubate at 4 °C for 15 min and then at 35 °C for 45 min. After incubation, the serum/virus mixes were removed from the cell monolayers and 200 µL of Opti-MEM (Gibco) supplemented with 1X antibiotic/antimycotic solution (Gibco) and 1 µg/mL of TPCK-Trypsin was added to each well. Plates were set to incubate for 72 h at 35 °C, under 5% CO_2_. Virus neutralization titers were determined by HA assay. The limit of detection was at dilution of 1/10. Samples with undetectable VN titers were assigned a dilution value of 1/8 for statistical purposes.

### 2.11. Microarray for IgG and IgA Determination

Sera collected at 19 dpb and 14 dpc, and nasal washed collected at 14 dpc were analyzed through protein microarrays to determine anti-HA, -NA, and -NP IgG and IgA levels from multiple Victoria- and Yamagata-like IBVs (Table 1). Purified IBV protein antigens were purchased from Sino Biological (Wayne, PA) (Table 1). Microarrays were carried out as described elsewhere [38]. The results are expressed as the group mean fluorescence intensity (MFI) ± SD. The higher the MFI, the more Abs bound to a particular antigen. MFIs were plotted using Prism v9 (GraphPad).

## 3. Results

### 3.1. FluB-RAM and FluB-RANS Viruses with Rearranged Genomes

The currently available influenza B virus LAIVs approved for human use are based on cold-adapted/temperature sensitive mutations. More recently, we developed an alternative influenza B virus LAIV based on amino acid mutations on the PB1 segment with or without a C-terminal HA tag. To further expand the choice of potential alternative LAIV candidates against the influenza B virus and to test the hypothesis that different LAIV backbones have an impact on adaptive immunity, we developed two strategies of genome re-arrangement within the backbone of the B/Brisbane/60/2008 strain (Victoria lineage). The first strategy, FluB-RAM, consists of moving the BM2 ORF from segment 7 into the C-terminal end of the PB1 ORF in segment 1 (Figure 1A). Segment 1 is further modified with the inclusion of a linker peptide sequence (G4S) and the Tav 2A protease sequence between the PB1 and BM2 ORFs. The strategy leads to a chimeric polymerase PB1 subunit protein carrying the G4S linker and the Tav 2A protein sequences and the BM2 protein, but with N-terminal proline. Segment 7 is mutagenized to eliminate the codon for the first methionine in the BM2 ORF, the inclusion of an additional stop codon in the BM1 ORF, and two early stop codons in the BM2 ORF, resulting in complete obliteration of BM2 expression from its cognate segment. In the second strategy, FluB-RANS (Figure 1A), the NS2 ORF from segment 8, instead of BM2 ORF is cloned downstream of PB1. Segment 8 is further modified to produce an amino acid mutation and a stop codon at the splicing acceptor boundary and an additional stop codon in BNS2 (F38Stop). In addition, codons 15, 18, and 31 were mutated (M15T, M18T, and M31T), to prevent leaky expression of a truncated BNS2 protein. The FluB-RAM and FluB-RANS viruses were successfully rescued and propagated in ECEs. To quickly visualize whether the viruses contained the corresponding rearranged PB1 gene segments, RT-PCR targeting the region containing the BM2 or BNS2 insertions was performed (Figure 1B). The RT-PCR showed that the FluB-RAM and FluB-RANS viruses carry PB1 segments with the expected size changes (402 and 444 base pairs, respectively). The sizes of the amplified fragments were consistent with those of the positive control reverse genetics plasmids used to generate the corresponding viruses (Figure 1B). Further, the sequencing results confirmed the presence of the BM2 and BNS2 inserts downstream of the PB1 ORF in the corresponding viruses, as well as the mutations introduced in segments 7 and 8 that prevent the expression of BM2 or BNS2, respectively (Table 2). To further evaluate genome re-arrangement stability and the mutations introduced in the M and NS gene segments, five serial passages from an E1 stock were performed in ECEs, as described above. Segments 1 and 7 from the FluB-RAM virus and 1 and 8 from the FluB-RANS virus from each passage were amplified by RT-PCR and sequenced by Sanger (Table 2). In addition, NGS was performed on the last passage virus and compared with the original stock virus from passage 1. Both Sanger sequencing and NGS confirmed the presence of the BM2 or BNS2 downstream of the PB1 gene segment in either FluB-RAM or FluB-RANS, respectively. The sequencing results also confirmed the maintenance of the mutations introduced in either the M or NS gene segment from the corresponding virus. These results highlight the stability of the two genome re-arrangement strategies introduced in the B/Bris genome.

### 3.2. FluB-RAM and FluB-RANS Viruses Are Attenuated In Vitro

To determine the growth of the rearranged viruses at different temperatures, MDCK cells were infected with either B/Bris WT, FluB-RAM, or FluB-RANS at 0.01 MOI. The growth kinetics for each virus was assessed at 33 °C, 35 °C, and 37 °C for up to 96 hpi (Figure 1C). Compared with the B/Bris WT virus, both FluB-RAM and FluB-RANS showed significantly lower replication at all three temperatures. Of note, the replication of the FluB-RANS virus was lower than that of the FluB-RAM virus at either 33 °C or 35 °C and was almost undetectable at 37 °C compared with the B/Bris WT and FluB-RAM viruses. These results demonstrate that both FluB-RAM and FluB-RANS are attenuated in vitro. Rearranged virus yield in ECEs reached titers of 1 × 10^8^ and 3.16 × 10^8^ EID_50_/mL for FluB-RAM and FluB-RANS, respectively.

### 3.3. FluB-RAM and FluB-RANS Viruses Show Differences in Attenuation

The safety and immunogenicity of the FluB-RAM and FluB-RANS viruses were tested in DBA/2J mice, a small animal model susceptible to influenza B viruses without further adaptation [26]. DBA/2J mice (7-week-old, male and female) were inoculated with 10^6^ EID_50_/mouse i.n. following a prime/boost strategy 20 days apart with the corresponding rearranged virus (Figure 2A). As a control, a group of mice was inoculated with the B/Bris WT virus (10^6^ EID_50_/mouse i.n.). Prime vaccination with the FluB-RANS resulted in neither clinical signs nor body weight changes in both male and female mice (Figure 2B). In contrast, male mice primed with the FluB-RAM virus showed an average of ~10% body weight loss between 7 and 9 dpv, but started to recover from 10 dpc onwards, whereas female mice showed a slight drop in body weight (<5%) at 7 dpv and quickly recovered. Consistent with the presentation of clinical signs, no mortality was observed in mice that received the FluB-RANS virus or female mice inoculated with the FluB-RAM virus (not shown). One out of 6 male mice primed with the FluB-RAM virus had to be euthanized by 10 dpv (not shown). These observations contrast with those in the group primed with the B/Bris WT virus, where male and female mice showed body weight drops of ~20% and where 4 out 6 males and 1 out 6 females succumbed to the infection between 8 and 10 dpv (not shown). These results show that the FluB-RANS vaccine candidate is the safest between the two rearranged viruses and both viruses are attenuated in vivo compared with the B/Bris WT strain. As expected, boost vaccination resulted in neither clinical signs nor mortality in any of the groups (data not shown).

### 3.4. Qualitative Differences in Humoral Responses among Different Vaccine Groups

The humoral responses induced by the rearranged virus vaccines were analyzed utilizing serum samples obtained at 19 days post-boost (19 dpb) from a subset of 4 mice/group (2 males, 2 females, except in the FluB-RAM group with 1 male and 2 female serum samples). Please note that we included HI data from FluB att as it was part of the same study, although it was reported elsewhere [39]; these data were included for comparison purposes. Boost vaccination led to HI titers above 40, the predictive limit of protection (Figure 2C) with either rearranged virus. HI titers for the FluB-RAM group (80, 160, and 160 for each mouse, respectively) and for the FluB-RANS group (80, 80, 80, and 160 for each mouse, respectively) were lower than those obtained with the B/Bris att virus (160, 160, 320, and 320 for each mouse, respectively). In addition, we performed virus neutralizations assays against B/Bris (homologous) and B/Wis (heterologous). Sera from both FluB-RAM and FluB att showed similar mean neutralization titers. The neutralization titers induced by FluB-RANS sera were significantly lower than those for the FluB att (*p* = 0.0095). As expected, no virus neutralization activity was detected against B/Wis (Figure 2C). To further understand possible differences in serological responses, IgG and IgA antibodies were analyzed using a protein microarray consisting of 22 HA proteins and 2 NA proteins derived from influenza B viruses (IBVs), corresponding to the two major lineages (Victoria and Yamagata), as well as a single NP protein from a prototypic IBV. Approximately one-third of the HA proteins are displayed as full length, whereas the rest correspond to the HA1 region. The array also contains many influenza A proteins, including group 1 and group 2 HA subtypes, NA subtypes, NP, M1, NS1, and NS2, which served as internal controls. Details of the strain of origin, source of the protein, and presence or absence of epitope tags are provided upon request. Side by side comparisons of the three vaccine groups, FluB-RAM, FluB-RANS, and FluB att, revealed qualitative differences in humoral responses for both IgG and IgA. Analysis of the serum samples post-boost showed that the FluB-RANS and FluB att groups had significantly higher anti-B/Brisbane/60/2008-HA IgG responses than the FluB-RAM group (*p* = 0.0019 and *p* = 0.0064, respectively) (Figure 3A top). In addition, FluB att anti-HA1 IgG responses were significantly higher than those for FluB-RAM for HA1 from B/Victoria/02/1987 (*p* = 0.0332), B/Ohio/01/2005 (*p* = 0.0257), B/Massachusetts/03/2010 (*p* = 0.0434), and B/Wisconsin/02/2012 (*p* = 0.0335) (Figure 3A top). Analysis of all the anti-Victoria HA responses combined and comparison between groups confirmed that the FluB-RANS vaccine induced higher responses than the FluB-RAM vaccine (*p* < 0.0001) (Figure 3A bottom). When looking at the anti-Yamagata responses, FluB-RANS showed numerically higher anti-HA IgG responses that the other vaccine groups; however, none of those were statistically significant (*p* > 0.05) owing to the high variability between samples within the group (Figure 3B top). When the responses against all the Yamagata lineage HAs were combined, the FluB-RANS group had a significantly higher IgG response compared with the other vaccine groups (*p* < 0.0001 and *p* < 0.0001, respectively) (Figure 3B bottom). In contrast, anti-HA IgA responses were numerically higher for both IBV lineages in samples from the FluB att group, but not significantly different than the other groups (*p* > 0.05) (Figure 3C,D). Combining the responses against all the Victoria or Yamagata HA antigens, FluB att induced significantly higher IgA responses than FluB-RAM (*p* < 0.0001 and *p* < 0.0001, respectively) and FluB-RANS (*p* = 0.0021 and *p* = 0.0097, respectively) against both lineages (Figure 3C,D bottom). Interestingly, and despite showing the least attenuation, FluB-RAM samples showed the lowest levels of anti-HA IgG and IgA responses among the three vaccine groups (Figure 3).

Differences in serological responses against NA and NP were also observed (Figure 4). Anti-NA IgG and IgA showed a trend towards higher responses in samples from the FluB att group, although most of them were not statistically significant (*p* > 0.05). However, the FluB att vaccine induced a significantly higher anti-B/Phuket/3073/2013 IgG response than the other two groups (*p* = 0.005 and *p* = 0.0016, respectively). It was noted that the NA antigen derived from the B/Phuket/3073/2013 provided more reliable signals with low background noise (Figure 4A,B). In contrast, the NA antigen derived from B/Brisbane/60/2008 reacted poorly in the array when probing for IgG responses and provided a high background signal when probing for IgA responses. The trend of anti-NP IgG responses was also numerically higher in samples from the FluB att group (Figure 4C). Anti-NP IgA serum responses were low, except for the serum from one female in the FluB-RAM group, which clearly show reactivity well above background (Figure 4D). Interestingly, samples from the FluB-RANS and FluB-RAM groups had similar anti-NA and anti-NP responses, despite their differences in anti-HA responses.

### 3.5. FluB-RAM and FluB-RANS Effectively Protect Mice against Lethal IBV Challenge

Protection efficacy of the rearranged viruses was tested using a lethal challenge dose of 10^7^ EID_50_/mouse of B/Bris/PB2 F406Y strain, administered i.n. [26] 3 weeks after boost. Mice in the three vaccine groups (FluB att data included for comparison) were fully protected as no signs of disease and no mortality were observed (Figure 5A,B). In contrast, PBS-vaccinated/challenged mice showed severe body weight loss. Only one female (out of 8) and none of the male mice survived in the PBS-vaccinated/challenged group, consistent with previous studies [26,30].

### 3.6. Qualitative Differences in Humoral and Mucosal Responses among Different Vaccine Groups at 14 dpc

HI responses at 14 dpc were similar among the three vaccine groups, with a mean antibody titer increase of about 1 log_2_ compared with post-boost HI titers, particularly in samples from the rearranged vaccine groups (Figure 5C). No statistically significant differences were observed between vaccine groups with trends like those observed post-boost. With respect to the rearranged vaccine groups, the data showed better responses in female mice than in male mice. Further analyses of serum and nasal wash samples collected at 14 dpc revealed recall IgG and IgA anti-HA responses against both the Victoria- and Yamagata-lineage antigens (Figure 6). As expected, the reactivity of serum samples from all vaccine groups against Victoria lineage HA antigens was 1.5–2-fold higher than to those of the Yamagata lineage (Figure 6A,B). Interestingly, the HA1 antigen derived from B/Hong Kong/05/1972 (before the split of the two IBV lineages) reacted well with samples from all groups (Figure 6A), whereas the HA1 antigen from B/Florida/4/2006 and B/Utah/02/2007 (Yamagata lineage) shows the lowest reaction with the serum samples (Figure 6B). Of note, the full-length HA of B/Florida/4/2006 reacted well with samples from all three vaccine groups (Figure 6B). The pattern of anti-HA Victoria lineage serum IgA was similar among all vaccine groups, where differences in reactivity could be attributed to the different antigens in the array (Figure 6C). Post-challenge serum IgA responses against Yamagata-lineage HA antigens showed reactivity patterns attributed also to the different antigens, but trending towards better reactivity in samples from the FluB-RANS group (Figure 6D).

Perhaps the most striking differences in IgG and IgA profiles were observed in the NW samples (Figure 7). The trend of serum IgG, but not IgA, anti-HA responses at 19 dpb (Figure 3) translated similarly in the NW samples for both IgG and IgA responses (Figure 7). Thus, a trend of higher IgG and IgA anti-HA responses was observed for samples of the FluB-RANS group, whereas those from the FluB-RAM and FluB att had the lowest of such responses and were like each other. The FluB-RANS group displayed significantly higher general anti-Victoria IgG readings than both FluB-RAM and FluB att (*p* < 0.0001) (Figure 7A bottom), and a higher anti-Yamagata IgG response than FluB att (*p* = 0039). IgA responses detected in the NW material appeared to be more robust than the IgG responses. When comparing groups within the same HA antigen, the FluB-RANS group had significantly higher anti-Victoria responses than FluB-RAM and FluB att for B/Massachusetts/03/2010 (*p* = 0.0019 and *p* = 0.0156), B/Brisbane/60/2008 (*p* = 0.0047 and *p* = 0.0370), B/Malaysia/2506/2004 (*p* = 0.0294 vs. Flu Batt only), B/Wisconsin/02/2012-HA1 (*p* = 0.0021 and *p* = 0.0122), B/Massachusetts/03/2010-HA1 (*p* = 0.0006 and *p* = 0.0014), B/Brisbane/60/2008-HA1 (*p* = 0.0134 vs. FluB att only), and B/Ohio/01/2005-HA1 (*p* = 0.0153 and *p* = 0.0322). When comparing general anti-Victoria and -Yamagata responses, FluB-RANS had significantly higher IgA responses than the other two groups (Vic- *p* < 0.0001 and *p* < 0.0001*; Yam- p* < 0.0001 and *p* = 0.0201) (Figure 7C,D bottom). Further, signals were stronger for the full-length HAs and more prominent for the Victoria-lineage antigens compared with the Yamagata-lineage antigens, as expected (Figure 7A,D).

Differences were also observed at 14 dpc in the pattern of IgG and IgA serum and NW reactivity against the NA and NP antigens on the array (Figure 8). Anti-NA and anti-NP IgG serum responses at 14 dpc were similar among vaccine groups (Figure 8A), but were close to background against NA and low against NP in NW at 14 dpc (Figure 8B). The pattern of IgA serum and NW against NA and NP at 14 dpc (Figure 8C,D) followed the patterns observed at 19 dpb, despite their initial low signals (Figure 4B,D). Thus, samples from the FluB-RANS group tended to have the lowest IgA responses in both serum and NW samples, whereas those from the FluB-RAM group had the overall highest responses, particularly against NP, although not statistically different (*p* > 0.05). Overall, the pattern of anti-NA and anti-NP responses post-challenge showed opposite trends with respect to the anti-HA responses at 14 dpc (Figure 6, Figure 7 and Figure 8).

## 4. Discussion

Influenza virus genome re-arrangement is a viable alternative for the development LAIV vaccines. We previously showed such potential within the background of a H9N2 virus carrying full-length H9 and H5 HA proteins while maintaining a full set of the remaining viral proteins [31]. Like the approach followed in this study, the NS2 ORF from the H9N2 IAV was inserted downstream the PB1 gene, whereas the NS segment was modified to carry the NS1 ORF and a prototypic H5 HA ORF with a modified monobasic cleavage site. The H9N2/H5 virus showed successful protection against lethal highly pathogenic avian influenza A/Vietnam/1203/04 (H5N1) in mice and ferrets [31]. The same strategy was used to generate H9N2 viruses successfully expressing enhanced green fluorescent protein (eGFP) and secreted *Gaussia* luciferase (GLuc), as well as a 2009 prototypic H1N1 virus expressing GLuc [31,40]. Based on these previous studies, we developed the FluB-RAM and FluB-RANS LAIV candidates, with the exception that these viruses do not express foreign antigens. The FluB-RAM and FluB-RANS viruses remained stable after six serial passages in ECEs, as shown by RT-PCR and Sanger and NGS sequence analyses.

In vitro growth of FluB-RAM and FluB-RANS viruses was impaired under multiple temperature conditions in MDCK cells compared with the WT B/Bris strain. Of the two rearranged viruses, FluB-RANS grew at lower titers than FluB-RAM and its growth at 37 °C was barely over the limit of detection at 72 hpi (Figure 1C). However, both rearranged viruses reached titers of at least 10^8^ EID_50_/mL in ECEs that would make them suitable as vaccine candidates. The growth kinetics results were consistent with the observations during the in vivo safety assessment. Both FluB-RAM and FluB-RANS were attenuated in comparison with the B/Bris WT. The safety profile of FluB-RANS showed more attenuation than another LAIV candidate, FluB att, with four amino acid mutations in PB1 (E48K, K391E, E580G, and S660A), resulting in no noticeable signs of disease and no body weight changes. In contrast, the FluB-RAM virus induced some body weight loss, particularly in male mice, with one of those having to be euthanized. Nevertheless, the clinical signs induced by FluB-RAM inoculation in male mice were significantly lower, whereas they were almost nonexistent in female mice compared with those observed with the B/Bris WT strain (Figure 2B).

It is important to note that, during the process of testing safety and efficacy of the different vaccines, we observed biological sex as a variable for susceptibility to IBV. In our experience, male mice were more prone than female mice to developing more significant signs of disease and mortality upon IBV infection (Figure 2B and Figure 5A,B). In addition, female mice, but not male mice showed a biphasic curve of associated clinical signs after IBV challenge, with an initial phase of pronounced body weight loss, recovery close to initial body weight, and then a second phase of mild body weight loss before a second recovery phase (Figure 5A). Sex differences related to susceptibility to IAV have been extensively characterized [41,42]. However, previous studies have determined that female mice are more susceptible than males to IAV infection. In this regard, the differences in the susceptibility of male versus female mice infected with IBV of this report follow the pattern observed in humans where biological males are more prone to hospitalization due to influenza than biological females. In addition, although non-statistically significant, antibody titers after boost vaccination and after challenge showed a trend towards higher responses in female than in male mice (Figure 2 and Figure 5 and data not shown). These observations are consistent with previous studies assessing the response to vaccination in humans and mice that revealed higher antibody responses, higher B cell responses, higher cross-reactive antibodies, and higher CD4+ T cell numbers in females compared with males [41,42,43,44,45]. Thus, understanding sex as a variable to study IBV susceptibility and vaccine responses is warranted, but beyond the scope of this report.

Comparing the results from the present study to previous observations with the FluB att virus published elsewhere, but part of the same study, the rearranged FluB-RAM and FluB-RANS viruses induced comparable HI antibody levels within 1 log2 difference of each other, both before and after challenge (Figure 2 and Figure 5). Further, qualitative differences in IgG and IgA responses were observed among different vaccine groups explored using a protein microarray (Figure 3 and Figure 4). It was interesting to observe that the most attenuated virus, FluB-RANS, led to overall higher anti-HA serum IgG responses before challenge. In contrast, anti-HA serum IgG responses from the FluB-RAM were among the weakest before challenge despite the virus being the least attenuated. The pattern of anti-HA serum IgG in samples from the FluB att was intermediate between the two rearranged vaccine groups. However, serum samples from the FluB att group were consistently higher for IgA against HA and for IgG and IgA against NA and NP antigens in the array. Further, overall IgG and IgA responses against NA and NP from the FluB-RANS group were among the weakest. Despite these qualitative differences, both rearranged viruses protected mice against lethal challenge with the B/Bris/PB2-F406Y strain. NW antibody responses after challenge were of particular interest as they reflect recall antibody responses to the site that would most efficiently prevent infection. In NW samples from the FluB-RANS group, anti-HA IgG and IgA responses were particularly prominent, but anti-NA/NP IgA responses were the weakest compared with other groups. Interestingly, anti-NA/NP IgA responses were more prominent after challenge in serum and NW samples from the FluB-RAM group. Thus, we observed opposite patterns between anti-HA and anti-NA/NP responses for the FluB-RANS and FluB-RAM groups and intermediate patterns for the FluB att group. These observations are significant because they suggest that humoral responses against different IBV antigens are not equally impacted by the different LAIV backbones. Despite the relatively lower attenuation of the FluB-RAM virus, it could be useful in dose sparing situations and/or in the presence of pre-existing immunity as complement boost vaccine. Previous studies have suggested that priming with an LAIV followed by a killed virus vaccine leads to more complete protective responses than prime-boost strategies using a single type of vaccine against the 2009 pandemic H1N1 virus. More relevant to this report, vaccination with a seasonal H1 LAIV (pre-2009 H1N1 antigen) followed by a boost with a pandemic H1 LAIV led to more robust protective responses than either vaccine administered twice [46,47]. Thus, it is tempting to speculate that one or more LAIV platforms could be used in prime-boost approaches that would improve the protective response of currently approved vaccines against IBV.

## 5. Conclusions

In this report, we successfully generated and assessed the efficacy of two LAIV IBV vaccines using genome re-arrangement. FluB-RAM and FluB-RANS are attenuated in vitro and in vivo. Regardless of differences in attenuation profiles, both LAIV vaccine candidates induced protective antibody responses, and effectively protected mice against lethal challenge. These results warrant more in-depth assessment of the FluB-RAM and FluB-RANS LAIVs in mice and other animal models, to further characterize their protection efficacy and the stimulation of different arms of the immune response. Biological sex-driven differences in responses to vaccination are to be further evaluated.

## Figures and Tables

**Figure 1 vaccines-09-00897-f001:**
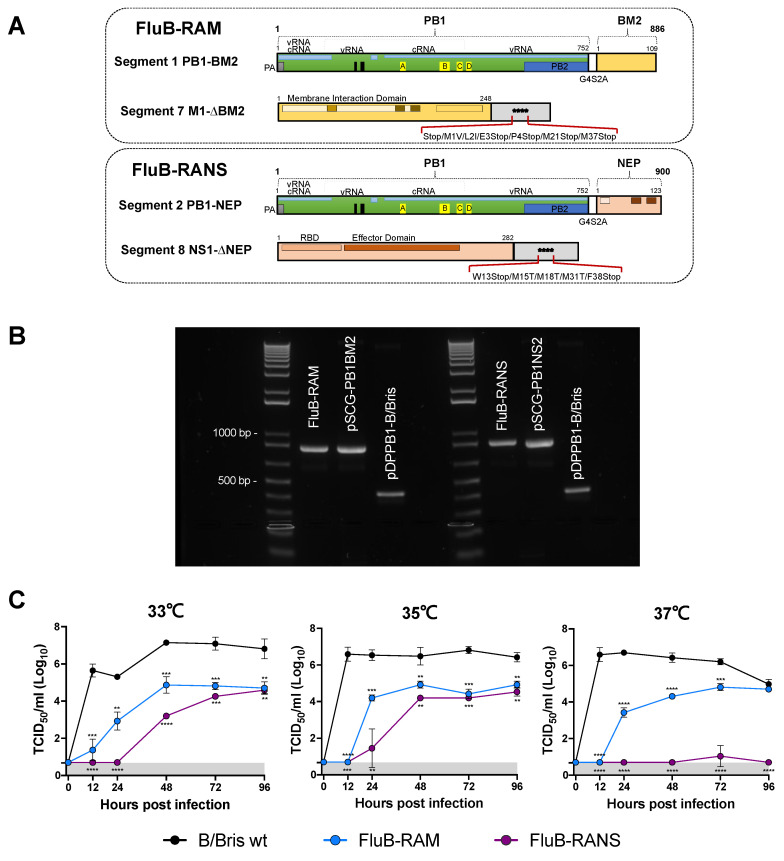
(**A**) Schematic representation of the modified PB1 and M or NS segments carried by the FluB-RAM (top) and FluB-RANS (bottom) viruses. (**B**) RT-PCR from newly rescued FluB-RAM and FluB-RANS carrying the rearranged PB1 gene segment. RT-PCR was performed from RNA extracted from FluB-RAM and FluB-RANS to amplify their rearranged PB1 gene segments. The plasmid carrying the PB1 WT and the rearranged PB1 plasmids were included in the reactions as controls. The agarose gel image showing the RT-PCR products is a demonstration that both rearranged PB1 gene segments amplified from FluB-RAM and FluB-RANS RNA carry either the BM2 or the BNS2, as confirmed by the corresponding controls. (**C**) Comparative growth kinetics of B/Bris WT, FluB-RAM, and FluB-RANS. MDCK cells were inoculated with the three viruses at an MOI of 0.01. Infected cells were incubated at 33 °C, 35 °C, and 37 °C to assess viral growth at different temperatures over time. Samples were collected at 0, 12, 24, 48, and 96 h post-infection (hpi) and titrated by TCID_50_ in MDCK cells. Virus titers are graphed as the mean TCID_50_/mL ± SD. Samples with undetected virus titers were assigned the limit of detection value (0.699 TCID_50_/mL). Data analysis and graphs were prepared using Prism v9. Curves were analyzed using multiple *t*-tests followed by the Holm–Sidak method to correct for multiple comparisons. Significant differences from the WT B/Bris are denotated by stars (*). ** = *p* < 0.01, *** = *p* < 0.001, and **** = *p* < 0.0001.

**Figure 2 vaccines-09-00897-f002:**
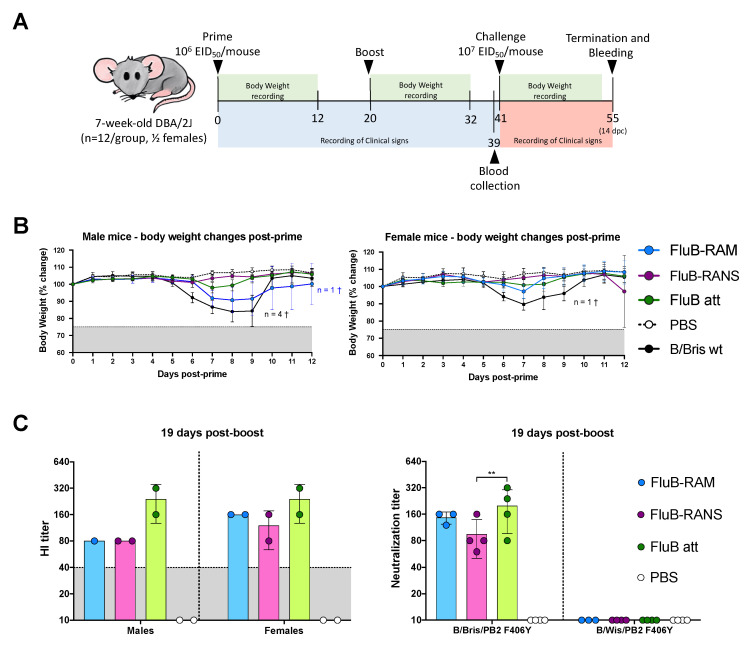
(**A**) Experimental timeline. Seven-week-old mice (*n* = 12/ group, ½ females) were vaccinated or mock vaccinated intranasally with 1× PBs, B/Bris WT, FluB-RAM, or FluB-RANS at day 0, monitoring body weight for up to 12 dpv and clinical signs for up to 20 dpv. Twenty days after vaccination (20 dpv), all mice were boosted with the same mock or vaccine treatment as before and their body weight was monitored for up to 12 dpb (day 32) and clinical signs were monitored for up to 21 dpb (day 41). At day 39 (19 dpb), a subset of mice was bled and humanely euthanized. The remining mice (*n* = 8/group, ½ females) were challenged at day 41 (21 dpb). Mice were observed for up to 14 dpc (day 55) for clinical signs and mortality, and body weight was recorded for up to 12 dpc (day 52). At 14 dpc, all remaining mice were bled and humanely euthanized; nasal washes (NW) were collected as well. (**B**) Monitoring of body weight and survival in male and female mice. After prime vaccination, body weight was monitored for up to 12 dpv; survival was recorded until the day before the boost. (**C**) HI antibody titers and VN titers after boost. Blood samples for serology were collected at 19 dpb. Sera were separated and used to perform HI assays comparing males and females, as well as to perform VN assays against B/Bris or B/Wis. Body weight values, HI titers, and VN titers were graphed as the group mean ± SD. Data analysis and graphs were prepared using Prism v9. Significant differences between groups are denotated by stars (*). ** = *p* < 0.01. † = number of dead mice.

**Figure 3 vaccines-09-00897-f003:**
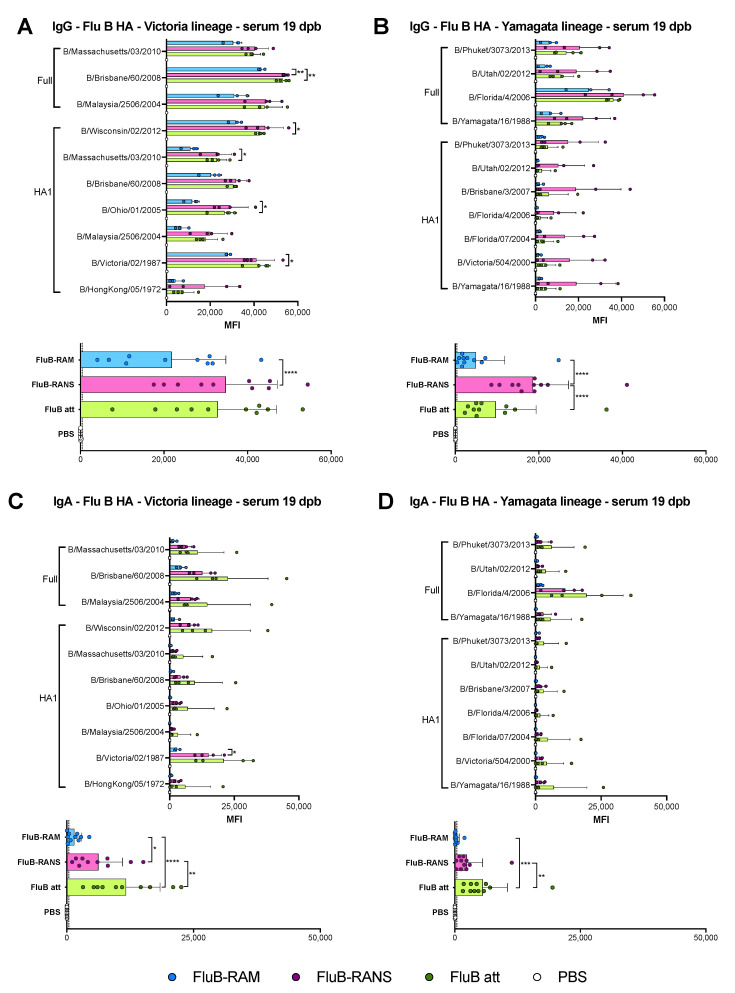
Lineage-specific IgG and IgA responses against the HA in serum at 19 dpb. IgG and IgA responses in serum were analyzed using protein microarrays. Sera collected at 19 dpb from mice inoculated with either PBS, FluB-RAM, FluB-RANS, or FluB att (n = 4/group) were tested against a variety of purified IBV full HA or HA1 portion protein antigens purchased from Sino Biological. The results are expressed as the group mean fluorescence intensity (MFI) ± SD. The higher the MFI, the more Abs bound to a particular antigen. (**A**) Anti-Victoria IgG responses. (**B**) Anti-Yamagata IgG response. (**C**) Anti-Victoria IgA responses. (**D**) Anti-Yamagata IgA responses. MFIs were plotted and analyzed using Prism v9. Top graphs from each subfigure show the responses from each group against every single HA protein antigen; the bottom graphs summarize the combined IgG or IgA responses against a particular lineage. Statistical analysis to compare responses between groups was performed using two-way ANOVA followed by a Tukey’s test for multiple comparisons. Significant differences between groups are denotated by stars (*). * = *p* < 0.05, ** = *p* < 0.01, *** = *p* < 0.001, and **** = *p* < 0.0001.

**Figure 4 vaccines-09-00897-f004:**
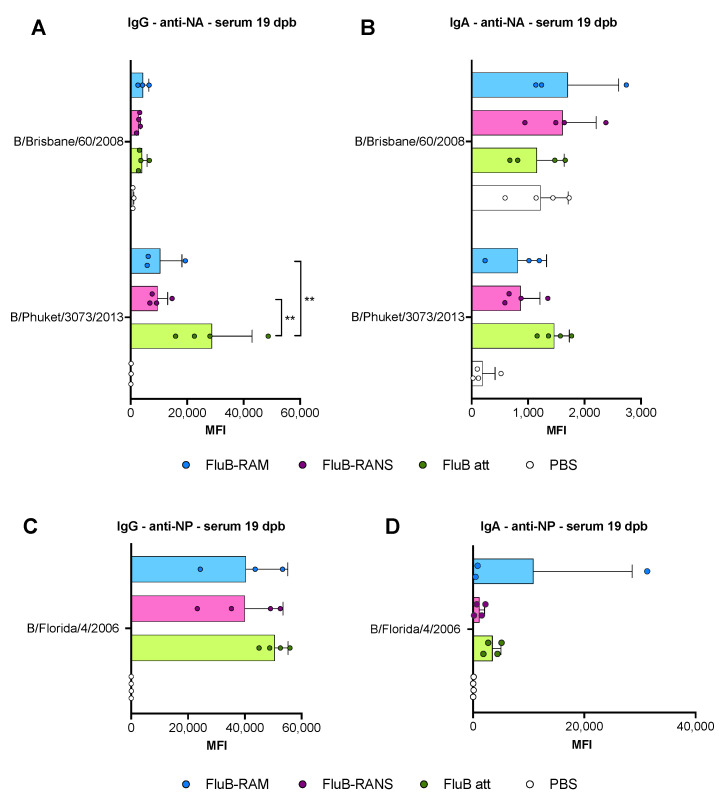
Lineage-specific IgG and IgA responses against the NA and the NP in serum at 19 dpb. IgG and IgA responses in serum were analyzed using protein microarrays. Sera collected at 19 dpb from mice inoculated with either PBS, FluB-RAM, FluB-RANS, or FluB att (n = 4/group) were tested against purified IBV NA or NP protein antigens purchased from Sino Biological. The results are expressed as the group mean fluorescence intensity (MFI) ± SD. The higher the MFI, the more Abs bound to a particular antigen. (**A**) Anti-NA IgG responses. (**B**) Anti-NA IgA responses. (**C**) Anti-NP IgG responses. (**D**) Anti-NP IgA responses. MFIs were plotted and analyzed using Prism v9. Statistical analysis to compare responses between groups was performed using two-way ANOVA followed by a Tukey’s test for multiple comparisons. Significant differences between groups are denotated by stars (*). ** = *p* < 0.01.

**Figure 5 vaccines-09-00897-f005:**
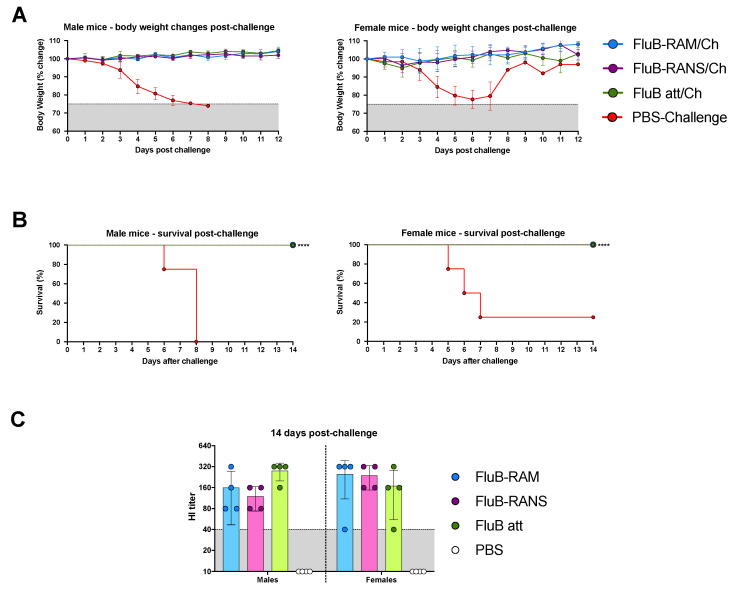
(**A**) Monitoring body weight in male and female mice. After challenge, body weight was monitored for up to 12 dpc. (**B**) Survival after challenge. Mortality in males and females was recorded until 14 dpc. (**C**) Post-challenge antibody titer determination. Blood samples were collected for serology at 14 dpc. Sera were separated and used to perform HI assays comparing males and females. Body weight values were graphed as the group mean ± SD. Survival data were analyzed using the log-rank test. HI titers are represented as the group mean ± SD. Data analysis and graphs were prepared using Prism v9. HI titers were compared between groups though a two-ay ANOVA followed by a Tukey’s test for multiple comparisons. Significant differences between group are denotated by stars (*). **** = *p* < 0.0001.

**Figure 6 vaccines-09-00897-f006:**
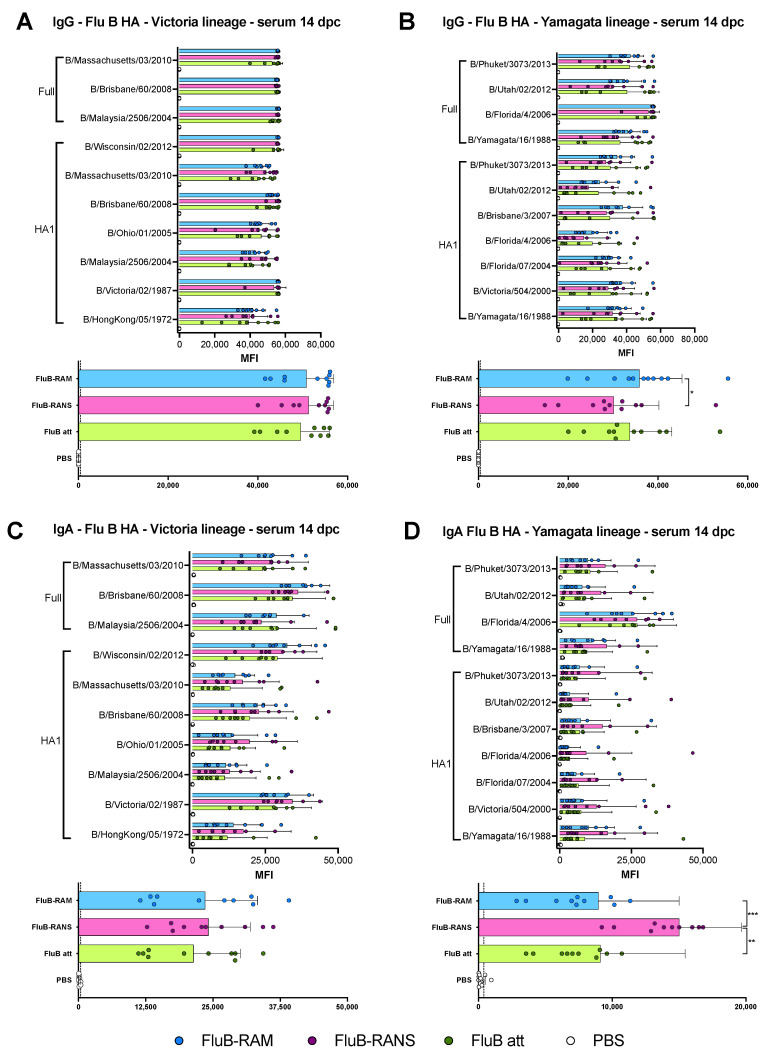
Lineage-specific IgG and IgA responses against the HA in serum at 14 dpc. IgG and IgA responses in serum after challenge were analyzed using protein microarrays. Sera collected at 14 dpc from mice challenged with B/Bris/PB2 F406Y (n = 4/group) were tested against a variety of purified IBV full HA or HA1 portion protein antigens purchased from Sino Biological. The results are expressed as the group mean fluorescence intensity (MFI) ± SD. The higher the MFI, the more Abs bound to a particular antigen. (**A**) Anti-Victoria IgG responses. (**B**) Anti-Yamagata IgG response. (**C**) Anti-Victoria IgA responses. (**D**) Anti-Yamagata IgA responses. MFIs were plotted and analyzed using Prism v9. Top graphs from each subfigure show the responses from each group against every single HA protein antigen; the bottom graphs summarize the combined IgG or IgA responses against a particular lineage. Statistical analysis to compare responses between groups was performed using two-way ANOVA followed by a Tukey’s test for multiple comparisons. Significant differences between groups are denotated by stars (*). * = *p* < 0.05, ** = *p* < 0.01, *** = *p* < 0.001.

**Figure 7 vaccines-09-00897-f007:**
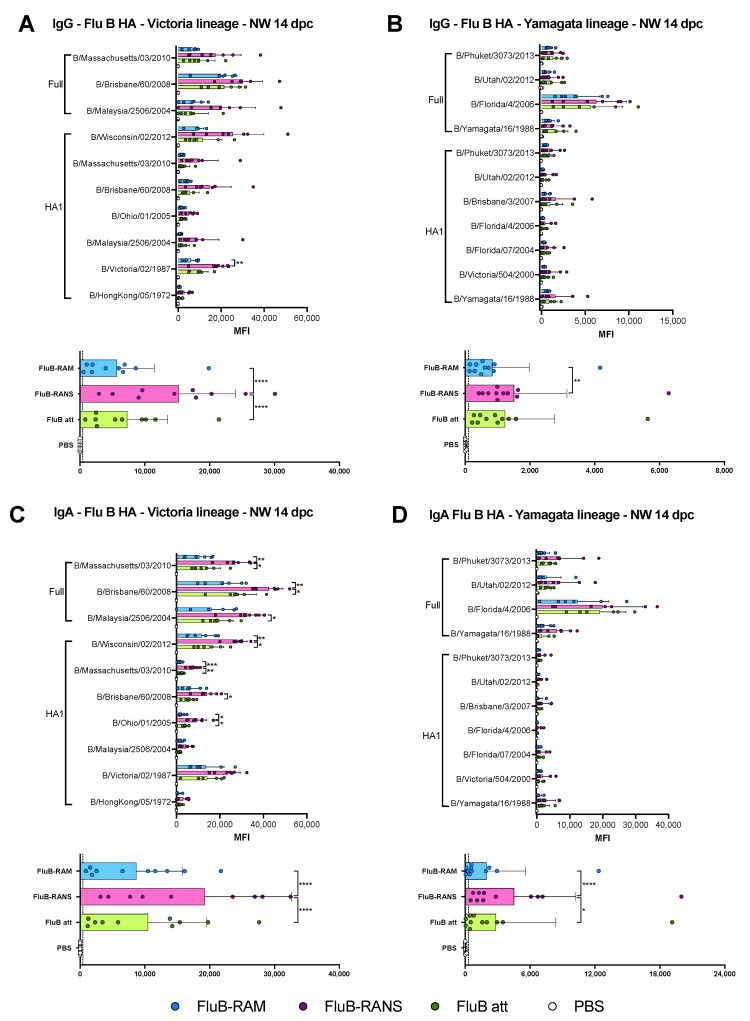
Post-challenge lineage-specific IgG and IgA responses against the HA in nasal wash (NW) material. IgG and IgA responses in NW were analyzed using protein microarrays. NW collected at 14 dpc from mice challenged with B/Bris/PB2 F406Y (n = 4/group) was tested against a variety of purified IBV full HA or HA1 portion protein antigens purchased from Sino Biological. The results are expressed as the group mean fluorescence intensity (MFI) ± SD. The higher the MFI, the more Abs bound to a particular antigen. (**A**) Anti-Victoria IgG responses. (**B**) Anti-Yamagata IgG response. (**C**) Anti-Victoria IgA responses. (**D**) Anti-Yamagata IgA responses. MFIs were plotted and analyzed using Prism v9. Top graphs from each subfigure show the responses from each group against every single HA protein antigen; the bottom graphs summarize the combined IgG or IgA responses against a particular lineage. Statistical analysis to compare responses between groups was performed using two-way ANOVA followed by a Tukey’s test for multiple comparisons. Significant differences between groups are denotated by stars (*). * = *p* < 0.05, ** = *p* < 0.01, *** = *p* < 0.001, and **** = *p* < 0.0001.

**Figure 8 vaccines-09-00897-f008:**
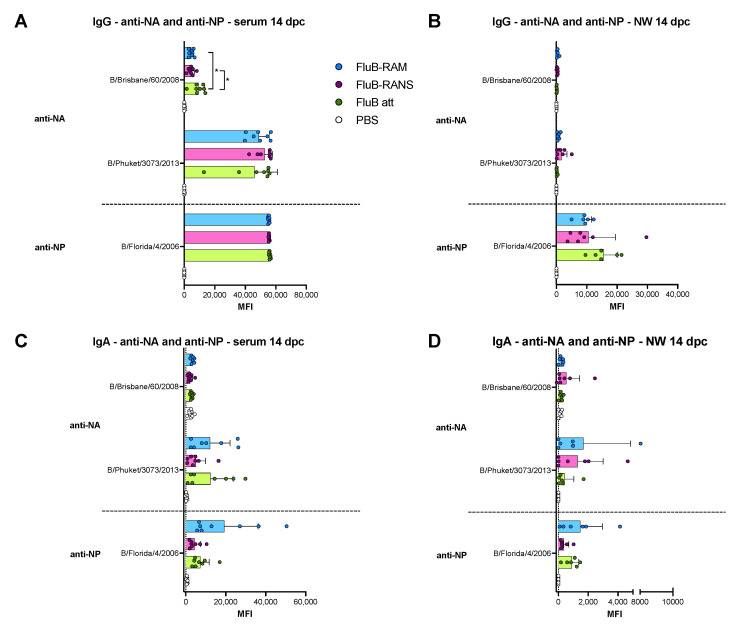
Post-challenge lineage-specific IgG and IgA responses against the NA and the NP in serum and NW. IgG and IgA responses in serum were analyzed using protein microarrays. Sera and NW collected at 14 dpc from mice challenged with B/Bris/PB2 F406Y (n = 4/group) were tested against purified IBV NA or NP protein antigens purchased from Sino Biological. The results are expressed as the group mean fluorescence intensity (MFI) ± SD. The higher the MFI, the more Abs bound to a particular antigen. (**A**) Anti-NA and -NP IgG responses in serum. (**B**) Anti-NA and -NP IgG responses in NW. (**C**) Anti-NA and -NP IgA responses in serum. (**D**) Anti-NA and -NP IgA responses in NW. MFIs were plotted and analyzed using Prism v9. Statistical analysis to compare responses between groups was performed using two-way ANOVA followed by a Tukey’s test for multiple comparisons. Significant differences between groups are denotated by stars (*). * = *p* < 0.05.

**Table 1 vaccines-09-00897-t001:** Protein antigens used in protein microarray analysis.

Protein	Region	IBV Strain	Lineage	Expression System	Catalog No.
HA	HA1	B/Victoria/02/1987	Victoria	HEK293	40163-V08H
HA	HA1	B/Wisconsin/01/2012	Victoria	HEK293	40462-V08H1
HA	HA1	B/Brisbane/60/2008	Victoria	HEK293	40016-V08H1
HA	HA1	B/Ohio/01/2005	Victoria	HEK293	40460-V08H1
HA	HA1	B/Massachusetts/03/2010	Victoria	HEK293	40191-V08H1
HA	HA1	B/Malaysia/2506/2004	Victoria	HEK293	11716-V08H1
HA	HA1	B/HongKong/05/1972	Victoria	HEK293	40461-V08H1
HA	HA1	B/Yamagata/16/1988	Yamagata	HEK293	40157-V08H1
HA	HA1	B/Victoria/504/2000	Yamagata	HEK293	40391-V08H
HA	HA1	B/Brisbane/3/2007	Yamagata	HEK293	40431-V08H1
HA	HA1	B/Phuket/3073/2013	Yamagata	HEK293	40498-V08H1
HA	HA1	B/Florida/07/2004	Yamagata	HEK293	40432-V08H1
HA	HA1	B/Utah/02/2012	Yamagata	HEK293	40463-V08H1
HA	HA1	B/Florida/4/2006	Yamagata	HEK293	11053-V08H1
HA	HA1+HA2	B/Brisbane/60/2008	Victoria	E. coli	40016-V08B
HA	HA1+HA2	B/Malaysia/2506/2004	Victoria	HEK293	11716-V08H
HA	HA1+HA2	B/Massachusetts/03/2010	Victoria	Baculovirus	40191-V08B
HA	HA1+HA2	B/Florida/4/2006	Yamagata	HEK293	11053-V08H
HA	HA1+HA2	B/Phuket/3073/2013	Yamagata	Baculovirus	40498-V08B
HA	HA1+HA2	B/Yamagata/16/1988	Yamagata	Baculovirus	40157-V08B
HA	HA1+HA2	B/Utah/02/2012	Yamagata	Baculovirus	40463-V08B
HA	HA1+HA2	B/Brisbane/60/2008	Victoria	HEK293	40016-V08H
NA	NA	B/Brisbane/60/2008	Victoria	HEK293	40203-VNAHC
NA	NA	B/Phuket/3073/2013	Yamagata	Baculovirus	40502-V07B
NP	NP	B/Florida/4/2006	Yamagata	Baculovirus	40438-V08B

**Table 2 vaccines-09-00897-t002:** Whole genome sequencing results after serial passages in embryonated chicken eggs.

	FluB-RAM		FluB-RANS	
Segment	Predicted Mutations	Egg Passage #6	Predicted Mutations	Egg Passage #6
PB1	+BM2 ORF	+BM2 ORF	+BNS2 ORF	+BNS2 ORF M495I (g1485a)
PB2	None	None	None	None
PA	None	None	None	S719P (t2184c)
HA	None	None	None	None
NP	None	None	None	A318T (g1012a)
NA	None	None	None	None
NB	None	None	None	None
BM1	Stop (c774a)	g543a ^s^ a546g ^s^ Stop (c774a)	None	None
BM2	M1V (a771g) L2I (c774a) E3Stop (g777a) P4Stop (c780t, c791g) M21Stop (a831t, t832a) M37Stop (a879t, t880a)	M1V (a771g) L2I (c774a) E3Stop (g777a) P4Stop (c780t, c791g) M21Stop (a831t, t832a) M37Stop (a879t, t880a)	None	None
NS1	None	None	None	None
NS2/NEP	None	None	a734t (Splicing acceptor) W13Stop (g737a) M14T (t743c) M15T (t752c) M31T (t791c) F38Stop (t812a c813a)	a734t (Splicing acceptor) W13Stop (g737a) M14T (t743c) M15T (t752c) M31T (t791c) F38Stop (t812a c813a)

^s^ Synonymous mutation. Higher case letters = amino acids. Lower case letters = nucleotides.

## Data Availability

The data discussed in this publication have been deposited in NCBI’s Gene Expression Omnibus [48] and are accessible through GEO Series accession number GSE180642.

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
