# Peer review of "FluB-RAM and FluB-RANS: Genome Rearrangement as Safe and Efficacious Live Attenuated Influenza B Virus Vaccines"

_vaccines, 2021, doi:10.3390/vaccines9080897_

Round 1

Reviewer 1 Report

Comments

The FluB-RAM and FluB-RANS: Genome re-arrangement as safe and efficacious live attenuated influenza B virus vaccines manuscript is a well-written and informative study showing the safety of two influenza B genome rearranged live attenuated vaccines.

Minor comments:

Line41-43 “The incidence of IBV infections varies from season to season, linked to 7-51% of the pediatric mortalities registered in the U.S. from 2004-2019

The percentage needs to be revised. i.e. Season 2009-2010 the pediatric mortalities registered in the U.S. due to influenza B are only 2 case and the total deaths due to influenza were 288 (0.69%)

The web page references need to include the links to this information.

Line 47-49:

The IBV linked pediatric mortalities cases are 122 cases, not 116 cases.

You should include the 2020-2021 cases report till now.

The paragraph showing the increased incidence of the IBV infection and mortalities will be more informative if you include a table or a chart showing this information.

Line 278:

Figure 2. figure legend (C) the word boost is missing the letter t.

Major comments:

  • The expression of the rearranged genes M and NS proteins in the mutant viruses need to be investigated compared to the IBV wild type (B/Bris wt) using Western blot for the MDCK infected cells.
  • The serum neutralization test; the gold standard humoral immunity test for evaluating the influenza virus vaccine protection was no evaluated with homologous or heterologous strains.
  • Although the live attenuated vaccines are valuable tools to induce the cell-mediated immune response as an indication of the vaccine's heterologous protection, the cellular immune responses were not evaluated in this study.

Author Response

Reviewer #1

The FluB-RAM and FluB-RANS: Genome re-arrangement as safe and efficacious live attenuated influenza B virus vaccines manuscript is a well-written and informative study showing the safety of two influenza B genome rearranged live attenuated vaccines.

Minor comments:

  • Line41-43 “The incidence of IBV infections varies from season to season, linked to 7-51% of the pediatric mortalities registered in the U.S. from 2004-2019”. The percentage needs to be revised. i.e. Season 2009-2010 the pediatric mortalities registered in the U.S. due to influenza B are only 2 case and the total deaths due to influenza were 288 (0.69%)

Thank you very much for pointing out this mistake. The percentages have been corrected and the range includes now seasons 2019-2020. Please see lines 42 and 43.

  • The web page references need to include the links to this information.

 The authors appreciate your input. The links to the mentioned references have been added, please see lines 744-749.

  • Line 47-49: The IBV linked pediatric mortalities cases are 122 cases, not 116 cases. You should include the 2020-2021 cases report till now. The paragraph showing the increased incidence of the IBV infection and mortalities will be more informative if you include a table or a chart showing this information.

 We apologize for the overlook; the numbers have been corrected accordingly.

  • Line 278: Figure 2. figure legend (C) the word boost is missing the letter t.

 It has been corrected accordingly. Thank you very much for pointing that out.

Major comments:

  • The expression of the rearranged genes M and NS proteins in the mutant viruses need to be investigated compared to the IBV wild type (B/Bris wt) using Western blot for the MDCK infected cells.

We appreciate your input and comments. We have confirmed that these viruses carry the corresponding gene segments by generation sequencing as reported in table 2. Since no issues were encountered during virus rescue, we assume that the corresponding proteins must be expressed as intended. We respectfully consider that the analysis of expression profiles of M or NS proteins from the vaccine candidates in comparison with the wild-type virus are out of the scope of this manuscript.

  • The serum neutralization test; the gold standard humoral immunity test for evaluating the influenza virus vaccine protection was no evaluated with homologous or heterologous strains.

Thank you very much for your comment. HI assay remains indeed as the gold standard assay to assess the antibody response and protection induced by vaccination. We have previously shown that antibodies with HI activity induced by vaccination with B/Bris (Victoria lineage) do not cross react with B/Wisconsin (Yamagata lineage), please see Santos et al (DOI: 10.1128/JVI.00056-17). To address your comment and to confirm our previous observations, we performed a virus neutralization assay against homologous and heterologous virus, please see figure 2-C.

  • Although the live attenuated vaccines are valuable tools to induce the cell-mediated immune response as an indication of the vaccine's heterologous protection, the cellular immune responses were not evaluated in this study.

We agree with your comment. We focused our analyses on humoral responses, which were comprehensively assessed with the antigen microarrays. Assessing the cell-mediated immune response is part of our plans for the further characterization of the vaccine candidates in mice and other species. We consider such assessment to be beyond the scope of the present report.

Reviewer 2 Report

Dear Authors,

Nicely performed study with clearly presented results and well discussed and interpteted. And thus it is suitable for publication in its present form.

Author Response

Dear Authors,

Nicely performed study with clearly presented results and well discussed and interpteted. And thus it is suitable for publication in its present form.

Thank very much, we appreciate your comments and the time invested on the review of this manuscript.

Round 2

Reviewer 1 Report

The manuscript has been improved in the revised version. The authors should consider the evaluation of the cell-mediated immune responses in future studies with attenuated live vaccines for influenza for their value in protection with the variant strains.